# Biomonitoring of benzophenones in guano samples of wild bats in Poland

**Slawomir Gonkowski[1], Julia Martín[2], Irene Aparicio[2], Juan Luis Santos[2], Esteban Alonso[2], Andrzej Pomianowski[3], László Könyves[4], Liliana Rytel[3]***

1 Department of Clinical Physiology, Faculty of Veterinary Medicine, University of Warmia and Mazury, Olsztyn, Poland, 2 Departamento de Química Analítica, Universidad de Sevilla, Sevilla, Spain, 3 Department of Internal Diseases with Clinic, Faculty of Veterinary Medicine, University of Warmia and Mazury in Olsztyn, Olsztyn, Poland, 4 Department of Animal Hygiene, Herd Health and Mobile Clinic, University of Veterinary Medicine, Budapest, Hungary

* liliana.rytel@uwm.edu.pl

**Data Availability Statement:** All data is included in the manuscript and the supplementary materials.

## Abstract

Benzophenones (BPs) are substances used in the production of sunscreens, cosmetics, and personal care products. However, there is a lack of knowledge of BPs in wild animals. Therefore, the study aimed to assess the concentration of selected BPs commonly used in the cosmetic industry in guano samples collected from 4 colonies of greater mouse-eared bats (*Myotis myotis*). Liquid chromatography with tandem mass spectrometry (LC-MS/MS) was used to determine guano concentrations of benzophenone 1 (BP-1), benzophenone 2 (BP-2), benzophenone 3 (BP-3) and benzophenone 8 (BP-8). BP-1 levels above the method quantification limit (MQL) were noted in 97.5% of samples and fluctuated from <0.1 ng/g to 259 ng/g (mean 41.50 ng/g, median 34.8). The second most common was BP-3, which fluctuated from <0.1 ng/g to 19 ng/g (mean 6.67 ng/g, median 5.05), and its levels higher than MQL were observed in 40% of samples. BP-2 and BP-8 concentrations did not exceed the method detection limit (0.04 ng/g) in any analyzed sample. There were visible differences in the BP-1 and BP-3 levels among the studied bat colonies. Mean BP-1 concentration fluctuated from 11.23±13.13 ng/g to 76.71±65.51 ng/g and differed significantly between the colonies. Mean BP-3 concentration fluctuated from 5.03±6.03 ng/g to 9.18±7.65 mg/g, but it did not differ significantly between the colonies. The results show that guano is a suitable matrix for the assessment of wildlife exposure to BPs. This could be particularly advantageous in protected species, where not disturbing and stressing the animals are crucial.

## 1. Introduction

Benzophenones (BPs) are a group of substances consisting of two phenyl groups linked to a carbonyl group [1]. Some BPs are of natural origin and are mainly produced by fungi and plants [2]. Some of them are synthesized for various branches of industry with an estimated annual production of 10,000 tons [3]. BPs absorb ultraviolet (UV) radiation, and therefore, they are foremost used to produce sunscreens with UV filters, cosmetics, flavorings, perfumes, hair dyes, shampoos, and personal care products [4, 5]. Moreover, BPs are part of food

**Funding:** Funded by the Minister of Science under the Regional Initiative of Excellence Program.

**Competing interests:** The authors have declared that no competing interests exist.

containers, bottles, paints, inks, plastics, car polishes, fabrics, and many other items, where they act as UV stabilizers [1, 6, 7]. There are 12 main benzophenone derivatives however, benzophenone 1 (BP-1, 4-dihydroxybenzophenone), benzophenone 2 (BP-2, 2,2',4,4'-tetrahydroxybenzophenone), benzophenone 3 (BP-3, 2-hydroxy-4-methoxybenzophenone, oxybenzone) and benzophenone 8 (BP-8, dioxybenzone) are the most common in industrial use (Fig 1) [3].

Additionally, the BPs levels in sunscreens and cosmetics, as well as in the environment, can increase significantly under favorable temperature and time conditions, probably due to the degradation of octocrylene—an organic compound commonly used in the beauty industry [5, 8].

Due to their widespread industrial use, BPs are increasingly reaching the natural environment. Such pollution by BPs is favored by their chemical properties, namely high stability, resistance to UV radiation, and ability to accumulate in water [9]. Previous studies have reported the presence of BPs, primarily in the inland surface and oceanic waters [10–12]. BPs also pollute tap water, air and soil, especially agricultural treated with compost from sewage sludge [13–15]. These substances may also penetrate edible for humans and animals plants [14, 16, 17]. BPs levels clearly depend on the urbanization and industrialization of the region, where the studies have been conducted, as well as on human activity in a given area [14, 18, 19].

The presence of BPs in various elements of the environment results in the exposure of humans and animals to these substances, which may get inside the living organisms through the digestive tract, skin, and respiratory system, as well as through the placenta during prenatal period [3, 20–23].

Exposure to BPs can cause disturbances in the functions of many internal organs since BPs may bind to receptors of various hormones, including estrogen, androgen, and thyroid hormone, and therefore show endocrine disrupting properties, which depend on the type of BP, dose, and animal species or type of cells [18]. Furthermore, BPs show genotoxic and mutagenic properties [24]. It is known that BPs affect male, and female reproductive systems, which manifests in perturbation in sperm development, decreased number of oocytes, prolonged estrus cycle, and disorders in the early germ cell development [24–26]. Some studies have also described the connections between BPs and infertility and the risk of endometriosis [27, 28]. In turn, BPs cause several changes within the nervous system, including i.e. alterations in the epigenetic state of neuronal cells, mitochondrial membrane potentials, and various levels of neuronal factors, which in turn can lead to abnormal brain development and an increased risk of neurodegenerative diseases [24, 29]. Some studies have found that BPs have obesogenic [30] and carcinogenic properties manifested by increased tumor cell proliferation and vascularity with simultaneous decrease in tumor cell apoptosis [31].

Due to abovementioned adverse effects of BPs, it is important to monitor the exposure to these substances in humans and animals. Many analytical methods have been developed to determine the concentration of these compounds in various matrices e.g. gas chromatography and high-performance liquid chromatography in combination with tandem mass spectrometry (GC-MS/MS, HPLC-MS/MS) being the most common [32, 33]. Other methods include high-performance liquid chromatography with ultraviolet spectroscopy (HPLC-UV), micellar electrokinetic chromatography with ultraviolet spectroscopy (MEKC-UV), ultra-high performance supercritical fluid chromatography- photo diode array detector (UHPSFC-PDA), low temperature plasma ionization mass spectrometry (LTP-MS), thin-layer chromatography with UV spectroscopy (TLC-UV), supercritical fluid chromatography with UV spectroscopy (SFC-UV), and gas chromatography-electron capture detector (GC-ECD) analysis [32, 34].

The vast majority of previous studies focused on the BPs exposure in humans, and the presence of these compounds has been confirmed in various sample types [35]. BPs have been

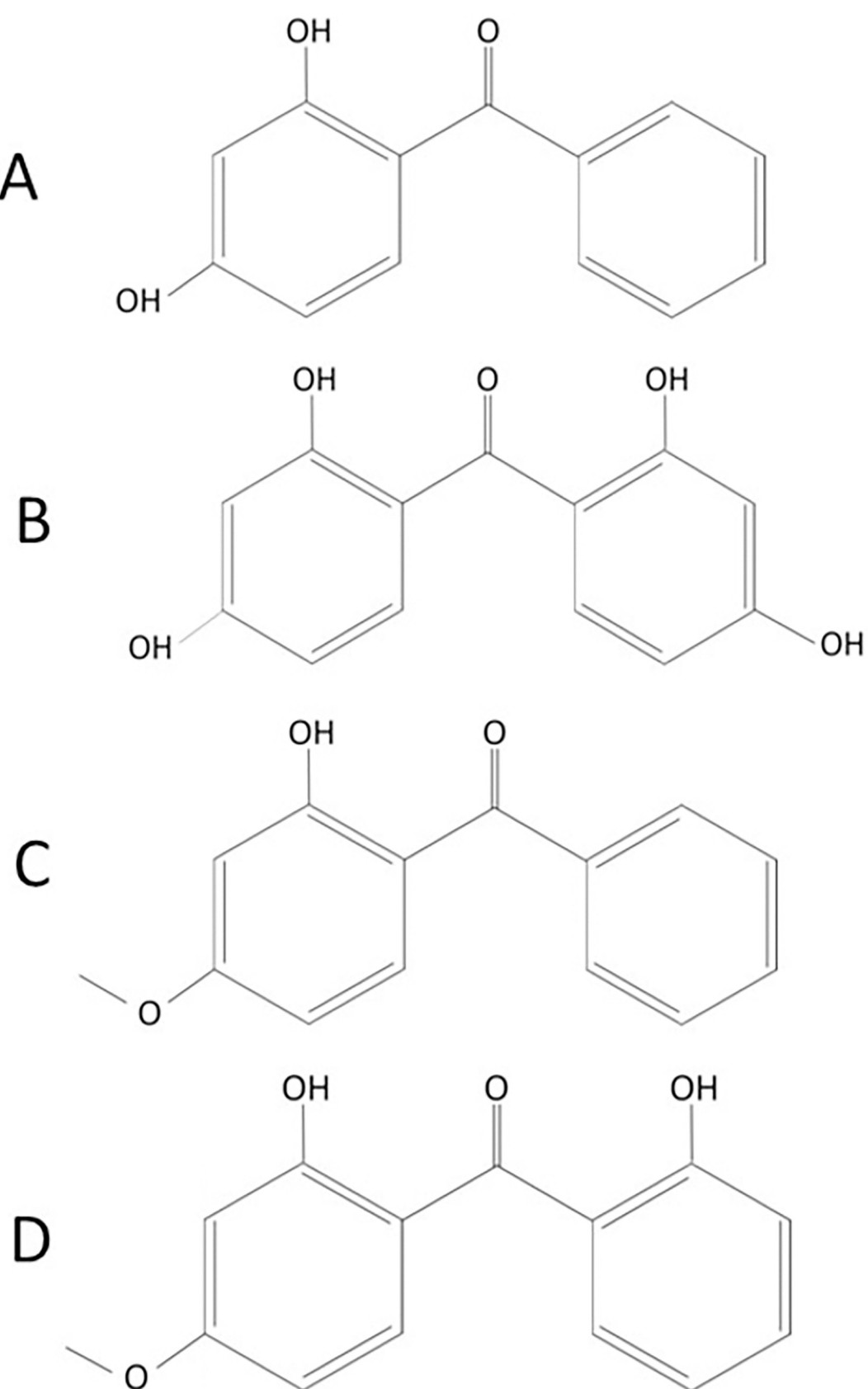

**Fig 1.** Structure of studied benzophenones (BPs): A) BP-1; B) BP-2; C) BP-3; D) BP-8.

described not only in typical matrices such as urine, [36, 37] whole blood, serum, and plasma [3, 38, 39], but also i.e. amniotic fluid [3], seminal plasma [39], placenta, [40] and breast milk [41]. Up to date research on humans have clearly shown that BPs levels depend on the part of the world, where investigations have been performed. Differences in BPs levels are not only linked to urbanization and industrialization, but also the type of work, habits and lifestyle, such as the frequency of cosmetic use [35, 42, 43]. Moreover BPs levels clearly depend on matrix used in analysis [37, 39, 41].

Contrary to humans, knowledge about wildlife exposure to BPs is relatively limited. Studies mainly describe the BPs concentrations in aquatic organisms because BPs, (especially BP-3), are particularly harmful to aquatic ecosystems due to their high water solubility and bioaccumulation potential [44–46]. Previous studies have shown that exposure to BP-3 results i.e. in various disturbances of the nervous system of zebrafish, which manifests in behavioral deficits learning and memory impairment, decreased neurogenesis, increased apoptosis within the brain and alteration in the enzymatic activity [47, 48]. Thus far the exposure to BPs have been described in wild marine mammals [49], turtles [50], fish [18, 51], mussels [52, 53], free-living flatworms [54], prawns, and urchins [55]. In addition, BPs have also been found in the eggs and internal organs of wild marine and terrestrial birds [56–58].

To the best knowledge of the authors, there are no studies on the exposure of terrestrial wild mammals to BPs with the exception of our earlier publication describing the methodology of examining bat guano for the presence of BPs on a very limited number of animals (n = 5) [59]. On the other hand, it is known that BPs have a strong harmful effects on terrestrial mammals. The majority of previous studies concerning this issue have been conducted on rodents and have shown that BPs affect blood hormone levels and hematological parameters, have adverse effect on the nervous system, and disrupt the functioning of the female and male reproductive systems [60–62].

However, it is very likely that terrestrial animals are exposed to BPs present in different elements of the natural environment. One group of animals, which play an important role in the ecotoxicology are bats. It is a large group of animals of over 1400 species inhabiting different climate zones [63]. Bats are highly sensitive to environmental pollution and environmental changes [64, 65], which is the reason for the continuous reduction of their population and the protection of many bat species [66–68]. Due to their sensitivity to environmental factors, bats are often used as a bioindicator of environmental quality during toxicological studies [69, 70]. Moreover, previous studies have shown that bats are highly exposed to various environmental pollutants, including e.g. heavy metals, organochlorine contaminants, parabens and pesticides, and bat guano is an appropriate matrix for such studies [71–73].

Matrix selection is a substantial problem in studies on wild animal exposure to environmental pollutants, when the levels of the investigated substances are determined. Obtaining vast majority of matrices used in toxicology e.g. urine, blood, or hair is practically impossible without restraining the animals, which is a strong stressor and a significant interference with their life and environment. This is particularly important in studies on protected species. Guano samples are practically the only matrix, which may be collected without significant interference [74, 75]. Moreover, our previous study has shown that guano samples can be used to determine BPs levels [59].

The above facts speak in favor that bats may be exposed and particularly sensitive to the harmful effects of BPs polluting the environment. Therefore this study aimed to answer the question whether and to what extent wild bats are exposed to BPs polluting the environment, as well as to determine whether the degree of this exposure depends on local factors. For this purpose the concentration levels of selected BPs commonly used in the industry (BP-1, BP-2, BP-3, BP-8) were evaluated in guano samples collected from the greater mouse-eared bats

(*Myotis myotis*) living in various parts of Poland. This study, to the best of our knowledge, is the first investigation on the exposure of terrestrial mammals to BPs and first to use oguano samples and will contribute to expanding knowledge on BPs in wild animals.

## 2. Materials and methods

### 2.1. Reagents

The following reagents were used: formic acid and ammonium acetate from Panreac (Barcelona, Spain); C18 disperse sorbent from Scharlab (Barcelona, Spain); HPLC-grade methanol, and water from Romil (Barcelona, Spain); Standards of benzophenone-1 (BP-1, purity: 99%), benzophene-2 (BP-2, purity: 97%), benzophenone-3 (BP-3, purity: 98%), benzophenone-8 (BP-8, purity: $\geq$98%) from Sigma-Aldrich (Steinheim, Germany). Isotopically labelled BP used as internal standard (BP-$d_{10}$) was purchased from Cambridge Isotope Laboratories (Tewksbury, MA, USA). Individual stock standard solutions were prepared at 1000 mg/L in methanol and kept at -18˚C. Working solutions were made up by dilution of the stock standard solution in methanol.

### 2.2. Sample collection

Bat guano samples were collected from August to September 2021 from four summer (nursery) colonies of greater mouse-eared bats (*Myotis myotis*) located in various regions of Poland (Fig 2).

The greater mouse-eared bat was chosen because it is the most common bat species in Poland despite being endangered species. This fact made sample collection easier, and such research can contribute to the protection of its population because the greater mouse-eared bat is especially sensitive to the environmental pollution with substances of human origin due to the fact that it establishes summer colonies in close proximity to human settlements.

The characterization of bat colonies included into the study is presented in Table 1.

Sample collection was performed as follows: flat glass containers were distributed throughout the floor where bat colony was located. After 48 hours, containers were removed and bat guano samples were placed in glass jars, frozen, and stored at -20˚C for further analysis. Ten guano samples collected from each bat colony were included in the investigation. Sample collection was carried out in a way that did not scare or stress the animals. Due to the non-invasive character of guano collection this study did not need the approval of the Local Ethics Committee for Animal Experiments based on the Polish law (the Act for the Protection of Animals for Scientific or Educational Purposes of 15 January 2015 Journal of Laws 2015, No. 266), Moreover, building administrators and personnel responsible for the protection of specific bat colonies gave verbal consent to collect samples. The samples were transferred to dark glass using sterile glass vanes, frozen at -22˚C and stored at this temperature until further analysis.

### 2.3. Sample treatment

Sample preparation was carried as previously described by Martin et al. [59]. Guano samples were lyophilized, homogenized and grounded into powder. Then 1g of each sample was weighed into a 12 mL glass tube and supplemented with 100 µL of a methanol solution of BP-$d_{10}$ (250 ng/mL). At that point sample was sonicated with 7 mL of methanol (0.5% v/v, formic acid) as extraction solvent in a bath for 5 mins and centrifuged for 5 mins (4050 $\times$ g). The extraction procedure was repeated three times, and the supernatants were combined.

To remove interferences from the matrix, a clean-up procedure based on dispersive solid phase extraction (d-SPE) was performed by adding 0.3 g of C18 sorbent. The tubes were

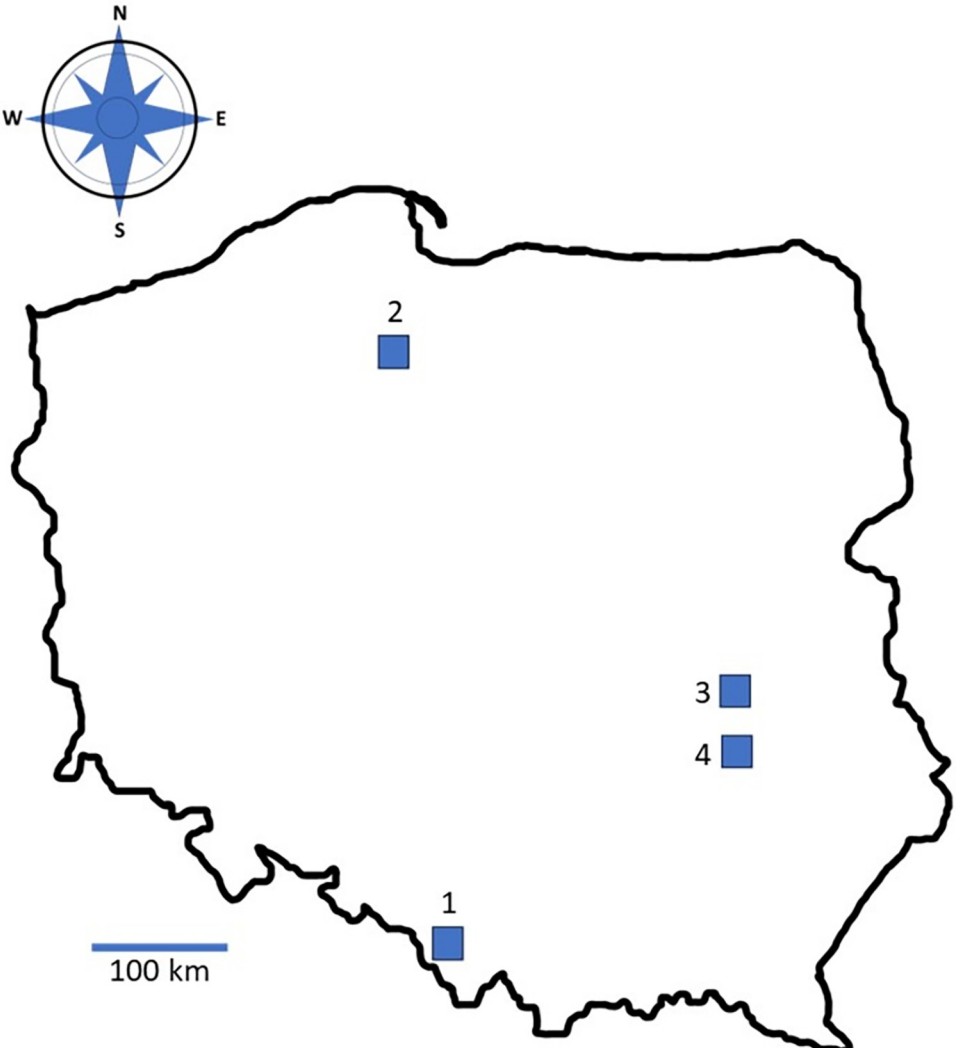

**Fig 2.** Localization of studied bat colonies 1) Brenna; 2) Śliwice; 3) Pulawy; 4) Opole Lubelskie.

shaken vigorously for 2 mins and then centrifugated at 4050× $g$ for 5 mins. The supernatant was evaporated under a nitrogen stream, reconstituted in 0.25 mL of a mixture methanol: water (50:50 v/v) until dry and filtered through a 0.22 μm nylon filter. A 10 μL aliquot was injected into the liquid chromatography tandem mass spectrometer (LC-MS/MS).

## 2.4. Liquid chromatography–tandem mass spectrometry conditions

Chromatographic analyses were performed using an Agilent 1260 Infinity II (Agilent, Santa Clara, CA, USA) as described by Martín et al. [59]. Separation was performed with a HALO C18 Rapid Resolution (50 x 4.6 mm i.d., 2.7 μm) (Teknokroma, Barcelona, Spain). The mobile phase consisted of methanol (solvent A) and a buffer solution acetic acid/ammonium acetate (pH 4.4) (solvent B). The elution program was as follows: 0–14 min, linear gradient from 28 to 70% of solvent A, increased to 80% of A in 5 min and to 100% of A in 6 min and held for 2 min. Flow rate was 0.6 mL/min.

**Table 1. Localization characteristics of studied bat colonies.**

| Bat colony No. | 1 | 2 | 3 | 4 |
|---|---|---|---|---|
| Name of place | Brenna | Sliwice | Pulawy | Opole Lubelskie |
| Type of place | village | village | city | town |
| Human population | 11300 | 2500 | 47400 | 8600 |
| Population density (per km$^2$) | 76.6 | 31.3 | 962.1 | 561.4 |
| Industry | none | none | Chemical, pharmaceutical, construction, food | food |
| Dominant landform | forest | agricularal | industrial | small-town |
| Straight line distance from the nearest large urban and industrial center (over 300,000 inhabitants) in km | 61 | 66 | 46 | 43 |
| Location of bat colony | School attic | Church tower | Children's home attic | School attic |
| Distance in the straight line from the nearest recreational water area/ beach/bathing beach (in km) | 3 | 4.7 | 2.2 | 0.6 |
| Approximate animal number in colony | 250 | 450 | 250 | 200 |
| coordinates | 49˚43'37.4"N 18˚53'46.3"E | 53˚42'19.0"N 18˚10'04.4"E | 51˚25'19.9"N 21˚58'37.4"E | 51˚09'02.7"N 21˚58'24.6"E |

Data according to statistical yearbooks published by the Central Statistical Office in Poland, https://stat.gov.pl/obszary-tematyczne/roczniki-statystyczne/

The LC system was coupled to a 6495 triple quadrupole mass spectrometer with electrospray ionization source operated in negative mode. Two multiple reaction monitoring (MRM) transitions, for identification and quantification purposes, were selected for each benzophenone. Instrument settings and analytical determination parameters are summarized in the Supplementary Material (S1 Table in S1 File).

## 2.5. Validation requirements

The analytical features (linearity, sensitivity and accuracy (trueness and precision)) of the method are presented in S2 Table in S1 File.

A matrix-matched calibration was performed for quantification purposes and to correct matrix effect. To this end, commercial guano samples were fortified with the analytes at eight different concentrations within the range from method quantitation limit (MQL) up to 100 ng/g dry weight—dw (0.1, 0.25, 0.5, 1.00, 5.00, 25.0, 50.0 and 100 ng/g dw). The commercial guano ("SuperGuano" from Top Crop - www.topcropfert.com) used for matrix-matched calibration was simultaneously analyzed. Commercial guano used in the study is 100% natural, collected in caves, where bats live far from the human influence. Note that the origin of the commercial guano used in matrix-matched comes from insectivores, like those analyzed in this study. BPs were not detected in the commercial guano samples.

To guarantee the quality of the obtained results, a protocol involving the use of control spiked samples including fortified commercial guano samples (25 ng/g dw), solvent, standards containing a mixture of the target compounds in pure solvent (100 ng/mL) and procedural blanks (without guano, processed the same way as the studied samples)were included into each analytical batch (5 samples).

## 2.6. Statistical analysis

The statistical analysis was performed using GraphPad Prism version 9.2.0 (GraphPad Software, San Diego, CA USA) and a Kruskal–Wallis test was used to determine the differences in BP-1 and BP-3 levels in selected bat colonies. Differences were considered statistically significant at $p < 0.05$.

**Table 2. Concentration (ng/g) and frequency of detection (%) of benzophenones in the analyzed guano samples (n = 40)—cumulative data.**

| Compound | Range (ng/g) | Arithmetic Mean | Geometric Mean | Median | Frequency of Detection above MQL (%) |
|---|---|---|---|---|---|
| BP-1 | <MQL-259 | 41.50 | 27.24 | 34.8 | 97.5 |
| BP-2 | <MDL | - | - | - | 0 |
| BP-3 | <MQL-19 | 6.67 | 3.83 | 5.05 | 40 |
| BP-8 | <MDL | - | - | - | 0 |

BP-1: benzophenone 1; BP-2: benzophenone 2; BP-3: benzophenone 3; BP-8: benzophenone 8; MQL—Method Quantification Limit ((0.10 ng/g dw); MDL—Method Detection Limit (0.04 ng/g dw)

## 3. Results

BP-1 was found in all samples included in the study (S3 Table in S1 File). Furthermore, its concentrations were the highest among the tested substances (Table 2).

BP-1 concentration fluctuated from <0.1 ng/g to 259 ng/g (mean 41.50 ng/g) and its concentration above MQL was observed in 97.5% of studied samples (S3 Table in S1 File). The second most common BP was BP-3, with concentration higher than MQL observed in 40% of the samples (Table 2). Its concentration levels fluctuated from <0.1 ng/g to 19 ng/g with a mean value of 6.67 ng/g. Noticeable differences in the BP-1 and BP-3 levels were observed even between samples collected from the same colony. Colony no.2 had the most prominent differences in BP-1 levels, which fluctuated from 30.5 ng/g to 259 ng/g (S3 Table in S1 File). Contrary to BP-1 and BP-3, the concentration of BP-2 and BP-8 did not exceed the method detection limit (MDL) (0.1 ng/g) in any studied sample (S3 Table in S1 File).

Furthermore, clear differences in BP-1 levels were found between studied bat colonies (Fig 3). Higher concentrations of this substance were found in bat colony no. 2 and no. 3, with the mean values (±SD)76.71±65.51 ng/g and 56.97±14.42 ng/g, respectively. In colonies no. 1 and no. 3 BP-1levels were statistically significantly lower than in colonies no. 2 and no. 3 and achieved 18.08±6.81 ng/g (colony no.1) and 11.23±13.13 ng/g (colony no. 4) (Fig 3A). Mean

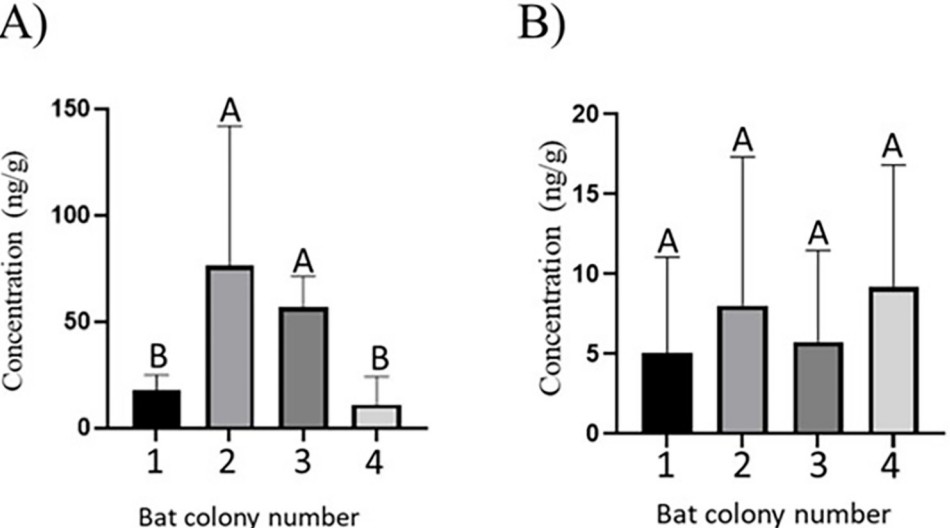

**Fig 3.** Mean concentration (±SD) of benzophenone 1 (BP-1) (A) and benzophenone 3 (BP-3) (B) in studied bat colonies. Statistically significant differences (p < 0.05) were marked with different letters, values not showing such differences—with the same letters. Bat colony in Brenna (1), Sliwice (2), Pulawy (3) and Opole Lubelskie (4). The figure was created using GraphPad Prism version 9.2.0 (GraphPad Software, San Diego, California USA).

**Table 3. Selected previous studies on concentration of benzophenone 1 (BP-1), benzophenone 2 (BP-2), benzophenone 3 (BP-3) and benzophenone 8 (BP-8) in humans and animals.** Benzophenone concentrations are shown in ng/g (in solid matrices) or ng/mL (in liquid matrices).

| Species | Country | Matrix | n | BP-1 | BP-2 | BP-3 | BP-8 | ref |
|---|---|---|---|---|---|---|---|---|
| Humans | Australia | urine | 24 | | | 16.5–312 | | [76] |
| | China | Amniotic fluid | 15 | <0.02–3.22 | | <0.01–0.38 | <0.01–0.60 | [3] |
| | Brazil | urine | 300 | 0.01–1910 | 0.25–8.25 | 0.7–9.83 | 0.01–2.69 | [36] |
| | China | urine | 100 | <0.07–14.6 | | <0.11–46.1 | | [77] |
| | | blood | 75 | <0.06–0.15 | | <0.41–3.88 | | |
| | Denmark | urine | 309 | <LOD-129 | <LOD-9.84 | <LOD-644 | | [39] |
| | | Blood serum | 306 | <LOD-0.86 | <LOD-0.10 | <LOD-5.18 | | |
| | | Seminal plasma | 297 | <LOD-10.9 | <LOD | <LOD-5.94 | | |
| | Spain | Breast milk | 10 | ND-0.51 | | ND-15.7 | ND-0.73 | [78] |
| | | placenta | 79 | <LOD-3.32 | 0.38–1.93 | 0.65–3.09 | <LOD | [79] |
| | | nails | 22 | 6.3–194 | ND | 25–245 | ND-3.9 | [80] |
| | USA | urine | 2517 | | | 0.4–21700 | | [42] |
| Dogs | USA | urine | 50 | <LOQ-9.2 | | <LOQ-46 | LOQ-0.7 | [81] |
| Cats | USA | urine | 50 | <LOQ-13.5 | | <LOQ-61.6 | <LOQ-8.4 | |
| Beluga whale | Canada | liver | 80 | | | <LOD-1320 | | [49] |
| Wild birds | Spain | eggs | | <LOQ-677 | | 16.9–49.3 | | [56] |
| Turtles | Italy | blood | 32 | | | <LOD-28430 | | [50] |
| Fish | Brazil | body | 11 | <LOQ | | 3.5–15.4 | | [82] |
| | Greece | body | 8 | ND | 11.8–41.9 | <LOQ-1.8 | | [83] |
| | Spain | body | 193 | | | ND-2.2 | | [84] |
| | Taiwan | body | 4 | 0.7–3.6 | | 3.3–6.9 | 0.5–2.4 | [85] |
| Wild bats | Poland | guano | 5 | 10.5–34.8 | <LOD | <LOQ-15.5 | <LOD | [59] |

LOD–limit of detection, LOQ-limit of quantification, ND- not detected

concentration levels of BP-3 amounted to 5.03±6.03 ng/g in colony no. 1, 8.02±9.31 ng/g in colony no. 2, 5.76 ng/g in colony no. 3 and 9.18±7.65 ng/g in colony no. 4 (Fig 3B). There were no statistically significant differences in BP-3 between colonies.

## 4. Discussion

Obtained results clearly indicate that wild bats are exposed to BPs. However, due to up to date research being conducted on humans and aquatic organisms, and using other matrices, the comparison of present results with previous investigations would be problematic to interpret (Table 3).

Nevertheless, our results confirm that bat guano is the appropriate matrix to evaluate the exposure of these animals to BPs, as we have observed in our preliminary research [59].

A vast majority of previous papers concern human exposure to BPs. Review article written by Mao et al. [35] mentions above 160 investigations describing BPs concentration levels in humans in various parts of the world mainly described BPs in the urine samples, but these substances have also been found in human blood, breast milk, amniotic fluid and even in the brain (Table 3). The latter proves the BPs ability to cross the blood-brain barrier. In turn, research on BPs in wildlife mainly focuses on fish and other aquatic animals, and only a few studies mention terrestrial birds (Table 3).

It is clear that the BPs concentrations in humans and animals vary significantly depending on the studied species, the region in which the study was conducted, and the type of BP. It is well known that BP levels in the living organisms depend on the presence of BPs in the natural

environment, resulting from the urbanization and industrialization of a given area, as well as the frequency of the use of sunscreens and personal care products by the studied region inhabitants [35, 42, 43].

Much less is known about interspecies differences in BPs metabolism. The knowledge of this is practically limited to selected BPs (mainly BP-3) and laboratory animals. Furthermore, the information is inconclusive. Some studies have shown differences in BP-3 metabolism between rats and mice [20, 60, 62, 86, 87], while other research has found that generally BP-3 metabolism in rats and mice is rather similar with only minor differences [88].

Given all the above, the comparison of the results obtained in the present study with previous investigations is rather difficult. However, this investigation has shown that all included bat colonies were exposed to BPs, which strongly suggests the environmental pollution with these substances in Poland. Furthermore, the results indicate that the greatest threat to wild bats are BP-1 and BP-3. It is in agreement with most previous studies, in which these compounds have been found in humans and animals in the highest amounts among BPs (Table 3). Relatively high BP-1 levels observed both in our and previous studies may result not only from high environmental pollution with this compound, but also from the BP-3 metabolism, because BP-1 can form in living organisms after BP-3 absorption [89].

Despite different matrices used in our and previous studies, it can be concluded that the exposure of wild bats to BP-1 and BP-3 is relatively high. Concentrations of BP-1 and BP-3 noted in our research are not only higher from those noted in terrestrial birds and most of marine organisms, but also from those noted in humans (Table 3). The latter fact is quite surprising, since BPs are anthropogenic pollutants, and humans are significantly exposed to BPs using sunscreens and personal care products containing these compounds [35]. Relatively high levels of BPs observed in guano samples of the greater mouse-eared bats noted in our study are most likely due to the lifestyle of this species. Namely, the greater mouse-eared bats establish the summer (nursery) colonies in the immediate vicinity of human settlements, usually in large attics and church towers [66]. Bats live in these colonies from March to October and they give birth and raise their offspring there. Due to being in close proximity to humans, they are highly exposed to anthropogenic pollutants, including BPs. The possibility of such contact and/or inhalation exposure to BPs is confirmed by previous studies that described the presence of benzophenones in indoor dust and air [21, 90]. The exposure of bats in the summer colony to BPs is also promoted by the use of these substances as ingredients of paints and wood impregnations [91, 92]. However, bat exposure to BPs through the gastrointestinal tract seems to be more important when guano samples are considered. The greater mouse-eared bats are insectivorous and feed on various arthropods (including i.e. carapid and other beetles, centipedes and spiders), that they glean from the ground [93]. To the best of our knowledge there are no studies on BPs levels in the terrestrial arthropods. However, BPs have been found in the soil and plants [14, 94], which allows us to assume that terrestrial invertebrates that constitute bat food may also contain these substances. Moreover, relatively high levels of BPs have been found in inland surface waters, that serves as a water source for bats [14, 95, 96].

However, it cannot be ruled out that higher concentration levels of BPs observed in our study compared to previous research result from the character of the matrix used. Namely, at least part of the BPs found in guano samples are substances that were present in water or food and were not absorbed through the digestive tract.

It should be pointed out that until now guano samples have not been used to assess human or animal exposure to BPs. It is because the main route of BPs elimination from living organisms is urinary excretion [20, 86, 88], and therefore most studies on humans have described BPs in urine samples [35]. However, it is known that BPs may also be eliminated with feces. Previous studies on [$C^{14}$]BP-3 in experimental rodents have found that 15% to 42% of this

substance (depending on the route of exposure and animal species) may be excreted with feces [85, 97]. For this reason, the analysis of guano samples seems to be an appropriate method to assess the degree of exposure of living organisms to BPs, which is confirmed by the results of our study. The usefulness of guano samples for determining bat exposure to BPs was also demonstrated in our preliminary research performed on very limited study population (n = 5) [59].

Guano analysis is particularly important for studies on wild protected animals. Contrary to collection of other types of samples (e.g. blood, urine, tissues, hair), that require animals to be captured or killed, feces samples may be gathered without significantly interfering with animal's life. Moreover, previous studies have indicated that bat guano may be an appropriate matrix to study the exposure of this animal group to various substances polluting the natural environment, such as parabens, pesticides and heavy metals [71–73]. Undoubtedly guano analysis also has its drawbacks such as impossibility to assign samples to certain animals, and therefore the animal's age, sex, and health status cannot be determined, and these parameters often affect the degree of exposure to environmental pollutants [98, 99]. On the other hand analysis of guano samples as an "aggregate sample" reflecting the degree of exposure of a whole bat colony can be very useful in research monitoring the impact of environmental pollutants on the bat population in a given area.

When guano samples are analyzed attention should be paid to collection of fresh samples, because old samples that remain outside for long periods of time are prone to either the decomposition of substances, or contamination with external substances present in the environment. Moreover, analysis of guano samples evaluates not only portion of the studied substance, which was absorbed into the body and then excreted with feces, but also portion that was not absorbed from consumed food and water. However, analysis of guano samples is a good alternative to "traditional" matrices in BPs biomonitoring in wild animals.

Statistically significant differences in BP-1 concentration in guano samples have been observed between bat colonies. It is in agreement with previous studies, which have found that BPs levels in the natural environment clearly depend on the region, where research has been conducted [35]. The differences noted in our study do not appear to depend only on the degree of industrialization and urbanization of the area. The second high concentration level of BP-1 was observed in colony no. 3 located in Pulawy—a relatively big city (47400 inhabitants) with chemical industry, however, the highest levels of BP-1 have been found in colony no. 2 located in a small village (2500 inhabitants) without any industrial plants. Therefore in this case, high levels of BP-1 in bat guano samples may be related to the use of BPs in agriculture [100]. Interestingly, clear differences in BPs levels were visible between colony no. 3 and no.4, although these colonies are located in close proximity, probably as a result of different urbanization and industrialization. Pulawy (colony no. 3, where BPs levels were higher) is the city with a large number of inhabitants, and a developed pharmaceutical and chemical industry. In turn, Opole Lubelskie (colony no. 4) is a much smaller town (8600 inhabitants) with poorly developed industry.

However, the obtained results have shown that bat exposure to BPs is affected by some unidentified factors that are difficult to determine without comprehensive environmental and demographic studies (e.g. studies on sunscreen, cosmetics, and personal care products consumption).

The question also arises, whether BPs levels noted in guano samples in our study may have a negative impact on bat health status. The answer is rather difficult. It is relatively well known that BPs affect the functions of many internal organs, influencing, i.e. the endocrine, reproductive, and neuronal systems. Furthermore, in studies conducted on wild animals, clear positive correlations between BP levels and other active sunscreen ingredients, gene biomarkers of

inflammation, oxidative stress, and hormonal activity have been found [50]. On the other hand, metabolism, toxicokinetic tissue distribution and therefore adverse effects of BPs depend on the animal species [20, 88], and until now these aspects in bats have not been studied. Due to the fact that untill now the evaluation of the degree of exposure to BPs through analysis of guano/feces samples have not been carried out in any animal species or in humans, the correlations between the levels of BPs in feces/guano and internal organs are unknown. Moreover it is known that BPs levels differ depending on the species and type of matrix studied (Table 3). Such differences may result from various metabolism of BPs in particular species and chemical properties of these compounds. It is commonly known that feces generally contain mainly oil-soluble substances, whereas urine contains more hydrophilic compounds. However, in spite of the fact that BPs are insoluble in water, the main matrix to studies on exposure to these substances is the urine due to the fact that the excretory system is the man rout of elimination of BPs from the living organism.

Given that BPs can also affect organisms at low doses [101], and wild animals are often exposed to a variety of synergistic environmental pollutants [102], it can be assumed that the levels of BP observed in this study may negatively affect bat health status. Nonetheless further research on this topic would be beneficial to confirm this hypothesis.

## 5. Conclusion

The present research is the first one to focus on the exposure of wild terrestrial mammals to BPs through the analysis of guano samples. Obtained results demonstrated that wild bats are exposed to BP-1 and BP-3, which confirms environmental contamination with these substances. These observations also strongly suggest that BPs polluting the environment may affect the health of wild terrestrial mammals and bats seem to be a good bioindicator for monitoring BPs levels in the environment.

Furthermore, our study has shown that despite some limitations guano samples are a good alternative to urine or blood samples in the assessment of the wild animal exposure to BPs. Such samples are of particular importance in research on protected wild animals, because collection of such samples is simple and does not require (unlike other matrices) capturing specific animals. However, due to intraspecies differences between BPs metabolism and toxicokinetics, and the lack of knowledge about the correlation between the presence of BPs in the guano and their distribution in internal organs, it is difficult to accurately assess the BPs levels in guano that are a sign of exposure to these substances harmful to the animal health. Therefore, further studies regarding this matter is crucial.

## Supporting information

**S1 File.**
(DOCX)

## Author Contributions

**Conceptualization:** Slawomir Gonkowski, Liliana Rytel.

**Investigation:** Liliana Rytel.

**Methodology:** Julia Martín, Irene Aparicio, Juan Luis Santos, Esteban Alonso.

**Writing – original draft:** Slawomir Gonkowski.

**Writing – review & editing:** Andrzej Pomianowski, László Könyves.

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
