## [Decision Letter · Decision Letter 0]

14 Feb 2024

PONE-D-23-43946Biomonitoring benzophenones in guano samples of wild bats in PolandPLOS ONE

Dear Dr. Rytel,

Thank you for submitting your manuscript to PLOS ONE. After careful consideration, we feel that it has merit but does not fully meet PLOS ONE’s publication criteria as it currently stands. Therefore, we invite you to submit a revised version of the manuscript that addresses the points raised during the review process.

We look forward to receiving your revised manuscript.

Kind regards,

Lyi Mingyang, Ph.D.

Academic Editor

PLOS ONE

Journal Requirements:

3. Please be informed that funding information should not appear in the Acknowledgments section or other areas of your manuscript. We will only publish funding information present in the Funding Statement section of the online submission form. Please remove any funding-related text from the manuscript. 

5. We note that Figure 1 in your submission contain map images which may be copyrighted. All PLOS content is published under the Creative Commons Attribution License (CC BY 4.0), which means that the manuscript, images, and Supporting Information files will be freely available online, and any third party is permitted to access, download, copy, distribute, and use these materials in any way, even commercially, with proper attribution. For these reasons, we cannot publish previously copyrighted maps or satellite images created using proprietary data, such as Google software (Google Maps, Street View, and Earth). For more information, see our copyright guidelines: http://journals.plos.org/plosone/s/licenses-and-copyright.

(1) You may seek permission from the original copyright holder of Figure 1 to publish the content specifically under the CC BY 4.0 license.  

6. Please upload a copy of Figure 3B, to which you refer in your text on page 14. If the figure is no longer to be included as part of the submission please remove all reference to it within the text.

Reviewers' comments:

Reviewer's Responses to Questions

**Comments to the Author**

1. Is the manuscript technically sound, and do the data support the conclusions?

Reviewer #1: Yes

Reviewer #2: Yes

Reviewer #3: Yes

2. Has the statistical analysis been performed appropriately and rigorously? 

Reviewer #1: Yes

Reviewer #2: Yes

Reviewer #3: Yes

3. Have the authors made all data underlying the findings in their manuscript fully available?

Reviewer #1: Yes

Reviewer #2: Yes

Reviewer #3: Yes

4. Is the manuscript presented in an intelligible fashion and written in standard English?

Reviewer #1: No

Reviewer #2: Yes

Reviewer #3: Yes

5. Review Comments to the Author

Reviewer #1: In the paper “Biomonitoring benzophenones in guano samples of wild bats in Poland Benzophenones in wild bats in Poland” the authors assessed the concentration levels of selected BPs commonly used in the industry including benzophenone 1 (BP-1), benzophenone 2 (BP-2), benzophenone 3 (BP-3) and benzophenone 8 (BP-8) in guano samples collected from colonies of greater mouse eared bats (Myotis myotis) using liquid chromatography with the tandem mass spectrometry (LC-MS/MS) method. This manuscript is well organized, and the drawn conclusions are coherent with the obtained results. I have enjoyed reading your paper; however, the manuscript must be revised by an English native speaker to fix several grammatical errors that I detected in the paper. I hope to provide very useful suggestions to improve the overall clarity of your study as well as the quality of your analysis. I think that my suggestions look feasible to you, and I believe you will be able to address them. Thus, please take care to do a full revision of your manuscript according to all my comments. Improvements based on my comments will be crucial for acceptance. I have some concerns and suggestions for each aspect of the manuscript. Please see below.

Abstract: I would like to suggest giving more emphasis to the results.

Line 30: To arrange the keywords in alphabetic order.

Introduction: The paper is technically sound and the claims are convincing. However I think that some references should be updated. Please, note that the hypothesis and the predictions are unclear, you need to well explain them.

Lines 98 - 99: I think that you should add these important references to support your sentence: “Bats are highly sensitive to environmental pollution and any environmental change”. I would like to suggest:

Salinas-Ramos, V. B., et al., (2021). Artificial illumination influences niche segregation in bats. Environmental Pollution, 284, 117187.

Huszarik, M., et al., (2023). Increased bat hunting at polluted streams suggests chemical exposure rather than prey shortage. Science of The Total Environment, 905, 167080.

Lines 99 - 100: I think that you should add these important references to support your sentence: “which is the reason for the continuous reduction in their population and the protection of many bat species”. I would like to suggest:

Ancillotto, L., et al., (2021). Resilient responses by bats to a severe wildfire: conservation implications. Animal Conservation, 24(3), 470-481.

Gilmour, L. R., et al., (2020). Comparing acoustic and radar deterrence methods as mitigation measures to reduce human-bat impacts and conservation conflicts. PLoS One, 15(2), e0228668.

Lines 109 - 110: I think that you should add these important references as example to support your sentence: “Feces/guano samples are practically the only matrix, which may be collected without the need for significant interference in the animal life. ”. I would like to suggest:

Del Vaglio, M. A., Nicolau, H., Bosso, L., & Russo, D. (2011). A first assessment of feeding habits in the fruit bat Rousettus aegyptiacus on Cyprus island. *Hystrix*, *22*(2), DOI:10.4404/hystrix-22.2-4587.

Provencher, J. F., Vermaire, J. C., Avery-Gomm, S., Braune, B. M., & Mallory, M. L. (2018). Garbage in guano? Microplastic debris found in faecal precursors of seabirds known to ingest plastics. *Science of the Total Environment*, *644*, 1477-1484.

Lines 112 – 117: Please, explain in detail you hypothesis and predictions.

Materials and methods: In general, the methods are appropriate and the study seems well conducted, although some details deserve a bit more attention i.e., especially about the methodology and the data. All the script used in this paper must be added in the supplementary materials. Please, provide also all the link to source where you downloaded the data.

Line 130: Why did you choose this bat species?

Figure 1: Please, add the scale and the north symbol.

Results: Well written! The figures and the tables are all informative and necessary, but not redundant, ensuring the correct comprehension of the manuscript.

Discussion: The paper discussed appropriately the context and the theme, although there is important literature not cited by the authors. I think that the authors should be discussing their results also comparing them with those already published on other species/genus/family. In fact your paper discusses findings in relation to some of the work in the field but ignores other important work that I think should be added in your discussion.

Reviewer #2: Dear Authors,

Congratulations on this interesting article. In general, I found the manuscript to be very well written, with substantial amount of data, with a clear structure, and the manuscript was easy to follow.

The authors analysed 40 bat guano samples from 4 sites in Poland for the content of benzophenones, using LC-MS/MS. The used LC-methodology was already published, but especially the analysis and results of the method used for these emerging pollutants will be of interest for readers of PLOS one. In my opinion the article is suitable for publication in PLOS one after Minor Revision.

Major remarks:

• L23/abstract: Using the median makes more sense as already used in Table 2

• L26-28: Rephrase the sentence. The results show that BPs can be determined by sampling guano and LC-MS/MS analysis and not to be an alternative to urine or blood samples. Blood samples were not compared to guano samples

• L32-117: I would suggest providing the structures of the BPs

• L32-117: Which other approaches are available for the analysis of BPs?

• L32-117: It should already be mentioned in the Introduction that the determined levels are compartment dependent as mentioned in the discussion

• L134: Rounding the population to hundreds is better, the exact number of citizens is irrelevant; e.g. 11,300

• L136-143: Good job!

• L176: Which commercial guano was used/ from which origin. I would assume there would be differences from the composition of the guano from insectivores, frugivores and nectarivores bats. The composition could have an influence on the sample preparation.

• L178: Why was 0.01 ng/g used as calibration point when the MDL is 0.04 ng/g. That makes no sense for me. Data should be reanalysed.

• L181-185: Good job!

• L196: Thank you for providing the single data in the supplementary material

• L271-277: I would assume that the used commercial guano is from a big frugivores bat colony (Madagascar?) and the colony is not next to a town/village/city. I a therefore not surprised that the commercial guano is not highly contaminated. It should be mentioned that Myotis myotis is an insectivore, feeding on various arthropods which could have an influence on the detected compounds.

• L278-290: Good point!

• L294-305: It is also an advantage that whole bat populations can be analysed by “one” sample. To my experience the difference of the pollutant load between individuals and the whole colony is not that great.

• L 320-335: I would assume there are differences in the distribution of the BPs in different compartments, and data must be available for other mammals (e.g. mice, humans). The faeces contain mainly fat-soluble substances whereas urine contains more hydrophilic substances

• L348-352: Good point!

• Figure 2: Should be revised. It is a bit confusing.

Minor remarks:

• L23: the concentration is already a level

• L32-117: The article of Schanzer et al could be useful (Chemosphere 2022, 135342)

• L121: disperse

• L122: What means high purity, please provide %

• L127: -18 space °C; no zero symbol

• L312/ L314: 47,400 and 2400

Reviewer #3: Studies on chemical pollutants in terrestrial environments are extremely important, especially emerging ones for which there is little or no information in the literature. Bats are mammals with unique characteristics, and therefore, they perform fundamental ecosystem services for the balance of the environment. Precisely performing these services and their proximity to agricultural and industrial areas, they become contaminated with the most diverse chemical pollutants.

Taking into account the difficulty in obtaining non-invasive samples to the individual, alternative matrices, such as the use of guano, are welcome.

The manuscript is easy to read and understand. I consider the article suitable for publication in “PLOS ONE”. However, some modifications must be taken into consideration.

Main concerns:

*Add information about benzophenone can increase significantly under favorable temperature and time conditions, probably due to the additional degradation of Octocrylene.

*L92: Add information about on the effects of benzophenone on laboratory rats.

*L131: Georeference the map.

*L134: Add information about recreational or tourist water areas. The use of UV-filter in these places is higher and can influence contamination.

*L220: Rephrase the figures, especially 2 and 3.

*The study needs to discuss more about the species’ characteristics, such as food guild, foraging areas and behavior, since the bioaccumulation rate can be affected by these topics. Furthermore, it is important to clarify some intraspecific information on the species based on the chosen matrix in the study.

Minor concerns:

*L62: Add information about carcinogenic potential.

*L87: Add current data on benzophenone-3 in zebrafish.

*L88: Add article "Comparative acute toxicity of benzophenone derivatives and bisphenol analogues in the Asian clam Corbicula fluminea."

*L88: Add article about freshwater planarian Dugesia japonica

*L144: As this is an unprecedented work, all details are important for future replications. Therefore, I suggest better developing how the samples were collected and stored in a way that prevents their accidental extrinsic pollution with BPs.

*L252: “Okereke and Abdel-Rhaman 1994; Okereke et al. 1994”

*L280: “Okereke”

6. PLOS authors have the option to publish the peer review history of their article (what does this mean?). If published, this will include your full peer review and any attached files.

Reviewer #1: No

Reviewer #2: No

Reviewer #3: **Yes: **Monteiro-Alves P.S

---

## [Author Response · Author response to Decision Letter 0]

19 Mar 2024

The authors thank for insightful reviews, that contributed to significant improvement of the manuscript. All comments of the Reviewers have been answered below (answers are marked in red font)

Reviewer #1: 

In the paper "Biomonitoring benzophenones in guano samples of wild bats in Poland Benzophenones in wild bats in Poland" the authors assessed the concentration levels of selected BPs commonly used in the industry including benzophenone 1 (BP-1), benzophenone 2 (BP-2), benzophenone 3 (BP-3) and benzophenone 8 (BP-8) in guano samples collected from colonies of greater mouse eared bats (Myotis myotis) using liquid chromatography with the tandem mass spectrometry (LC-MS/MS) method. This manuscript is well organized, and the drawn conclusions are coherent with the obtained results. I have enjoyed reading your paper; however, the manuscript must be revised by an English native speaker to fix several grammatical errors that I detected in the paper. I hope to provide very useful suggestions to improve the overall clarity of your study as well as the quality of your analysis. I think that my suggestions look feasible to you, and I believe you will be able to address them. Thus, please take care to do a full revision of your manuscript according to all my comments. Improvements based on my comments will be crucial for acceptance. I have some concerns and suggestions for each aspect of the manuscript. Please see below.

Answer: The authors thank the Reviewer for opinion. Manuscript has been improved by native speaker in English.

Abstract: I would like to suggest giving more emphasis to the results

Answer: The abstract has been reedited according the suggestion of the Reviewer. The results has been described in more detail (lines 23-32)

Line 30: To arrange the keywords in alphabetic order.

Answer: The keywords have been arranged in alphabetical order (line 34)

Introduction: The paper is technically sound and the claims are convincing. However I think that some references should be updated. Please, note that the hypothesis and the predictions are unclear, you need to well explain them.

Lines 98 - 99: I think that you should add these important references to support your sentence: "Bats are highly sensitive to environmental pollution and any environmental change". I would like to suggest: Salinas-Ramos, V. B., et al., (2021). Artificial illumination influences niche segregation in bats. Environmental Pollution, 284, 117187.

Huszarik, M., et al., (2023). Increased bat hunting at polluted streams suggests chemical exposure rather than prey shortage. Science of The Total Environment, 905, 167080.

Answer: References suggested by the Reviewer have been added (References no. 64 and 65)

Lines 99 - 100: I think that you should add these important references to support your sentence: "which is the reason for the continuous reduction in their population and the protection of many bat species". I would like to suggest:

Ancillotto, L., et al., (2021). Resilient responses by bats to a severe wildfire: conservation implications. Animal Conservation, 24(3), 470-481.

Gilmour, L. R., et al., (2020). Comparing acoustic and radar deterrence methods as mitigation measures to reduce human-bat impacts and conservation conflicts. PLoS One, 15(2), e0228668.

Answer: References suggested by the Reviewer have been added (references no. 67 and 68)

Lines 109 - 110: I think that you should add these important references as example to support your sentence: "Feces/guano samples are practically the only matrix, which may be collected without the need for significant interference in the animal life. ". I would like to suggest:

Del Vaglio, M. A., Nicolau, H., Bosso, L., & Russo, D. (2011). A first assessment of feeding habits in the fruit bat Rousettus aegyptiacus on Cyprus island. _Hystrix_, _22_(2), DOI:10.4404/hystrix-22.2-4587.

Provencher, J. F., Vermaire, J. C., Avery-Gomm, S., Braune, B. M., & Mallory, M. L. (2018). Garbage in guano? Microplastic debris found in faecal precursors of seabirds known to ingest plastics. _Science of the Total Environment_, _644_, 1477-1484.

Answer: References suggested by the Reviewer have been added (references no. 74 and 75)

Lines 112 - 117: Please, explain in detail you hypothesis and predictions.

Answer: This part of the manuscript has been reedited according to the suggestions of the Reviewer. Hypothesis and predictions have been explained in detail. (lines 144-148)

Materials and methods: In general, the methods are appropriate and the study seems well conducted, although some details deserve a bit more attention i.e., especially about the methodology and the data. All the script used in this paper must be added in the supplementary materials. Please, provide also all the link to source where you downloaded the data.

Answer: Some data in materials and methods have been added: description of commercial bat guano used in the study (lines 216-221), the manner of sample storage (lines 182-183), information about tourist waters around bat colonies in Table 1, source of information in Table 1 (line 169)

Line 130: Why did you choose this bat species?

Answer: Justification for the selection of bat species has been added (163-167)

Figure 1: Please, add the scale and the north symbol.

Answer : Figure 1 ( in new version Figure 2) has been reedited, the scale and north symbol have been added.

Results: Well written! The figures and the tables are all informative and necessary, but not redundant, ensuring the correct comprehension of the manuscript.

Answer: The authors thank the Reviewer for the opinion

Discussion: The paper discussed appropriately the context and the theme, although there is important literature not cited by the authors. I think that the authors should be discussing their results also comparing them with those already published on other species/genus/family. In fact your paper discusses findings in relation to some of the work in the field but ignores other important work that I think should be added in your discussion.

The authors thank the Reviewer for this suggestion. the authors tried to discuss the results in many ways. Of course, it is difficult to discuss all previous publications in detail, and too long discussion could make the article unreadable. The discussion has been supplemented with a description of the biology and nutrition of the bat species included into the study and possible relationships between lifestyle and the degree of exposure to BPs (lines 305-322). Moreover, information about previous studies on bat guano samples and character of guano as a matrix (lines 341-343, 346-349 and 383-393) and additionally description of bat colonies included into the study (lines 366-371) have been added.

Unfortunately, the Reviewer did not indicate which important previous publications on the topic described in the manuscript were omitted in the discussion. If the reviewer thinks that the additions are not sufficient, the authors ask to specify what matters should be described and what previous works should be taken into account

Reviewer #2: 

Dear Authors, Congratulations on this interesting article. In general, I found the manuscript to be very well written, with substantial amount of data, with a clear structure, and the manuscript was easy to follow. The authors analysed 40 bat guano samples from 4 sites in Poland for the content of benzophenones, using LC-MS/MS. The used LC-methodology was already published, but especially the analysis and results of the method used for these emerging pollutants will be of interest for readers of PLOS one. In my opinion the article is suitable for publication in PLOS one after Minor Revision.

Answer: Thank You very much for Your opinion

Major remarks:

* L23/abstract: Using the median makes more sense as already used in Table 2

Answer: Median values have been added according to the suggestion of the Reviewer (lines 23 and 24)

* L26-28: Rephrase the sentence. The results show that BPs can be determined by sampling guano and LC-MS/MS analysis and not to be an alternative to urine or blood samples. Blood samples were not compared to guano samples

Answer: Thank You for a good point. The sentence has been reedited (lines 30-32).

* L32-117: I would suggest providing the structures of the BPs

Answer: The figure (figure 1) with the structures of BPs has been added. Figure 1 caption – lines 48-49

* L32-117: Which other approaches are available for the analysis of BPs?

Answer: Other approaches of BPs analysis have been mentioned (82-91)

* L32-117: It should already be mentioned in the Introduction that the determined levels are compartment dependent as mentioned in the discussion

Answer: Information suggested by the Reviewer has been added to the introduction (lines 96-100)

* L134: Rounding the population to hundreds is better, the exact number of citizens is irrelevant; e.g. 11,300

Answer: Changes suggested by the Reviewer have been made (table 1 – line 169)

* L136-143: Good job!

Answer: Thank You

* L176: Which commercial guano was used/ from which origin. I would assume there would be differences from the composition of the guano from insectivores, frugivores and nectarivores bats. The composition could have an influence on the sample preparation.

Answer: Description of commercial guano has been added (lines 216-221)

* L178: Why was 0.01 ng/g used as calibration point when the MDL is 0.04 ng/g. That makes no sense for me. Data should be reanalysed.

Answer: The reviewer is right. It is a typographical error. The calibration curve was built from their MQL (0.10 ng/g dw) to 100 ng/g dw. The lowest point considered in the calibration curve for each compound was its MQL, since it is the lowest amount of analyte in a sample that can be quantitatively determined with suitable precision and accuracy. The manuscript was corrected (lines 213-216)

* L181-185: Good job!

Answer: Thank You

* L196: Thank you for providing the single data in the supplementary Material

Answer: Thank You for your opinion

* L271-277: I would assume that the used commercial guano is from a big frugivores bat colony (Madagascar?) and the colony is not next to a town/village/city. I a therefore not surprised that the commercial guano is not highly contaminated. It should be mentioned that Myotis myotis is an insectivore, feeding on various arthropods which could have an influence on the detected compounds.

Answer: Commercial guano has been described in materials and methods (lines 216-221). Behavior and feeding of Myotis Myotis have been described in the Discussion (lines 305-322)

* L278-290: Good point!

Answer: Thank You

* L294-305: It is also an advantage that whole bat populations can be analysed by "one" sample. To my experience the difference of the pollutant load between individuals and the whole colony is not that great.

Answer: the authors are in agreement with the Reviewer. Discussion has been supplanted according of the reviewer's suggestions (lines 346-349)

* L 320-335: I would assume there are differences in the distribution of the BPs in different compartments, and data must be available for other mammals (e.g. mice, humans). The faeces contain mainly fat-soluble substances whereas urine contains more hydrophilic substances.

Answer: Till now guano/feces of any species has not been used as a matrix to studies on BPs concentration. Additional information has been added into discussion according to the suggestions of the Reviewer (lines 384-393)

* L348-352: Good point!

Answer: Thank you

* Figure 2: Should be revised. It is a bit confusing.

Answer: : legend of figure (in old version figure 2, in new version figure 3) has been reedited (257-261)

Minor remarks:

* L23: the concentration is already a level

Answer: correction has been made (line 21)

* L32-117: The article of Schanzer et al could be useful (Chemosphere 2022, 135342)

Answer: the article suggested by the reviewer has been added (Reference no. 72)

* L121: disperse 

Answer: correction has been made (line 149)

* L122: What means high purity, please provide %

Answer: purity of the reagents has been added in % (lines 151-152)

* L127: -18 space °C; no zero symbol

Answer: correction has been made (line 155)

* L312/ L314: 47,400 and 2400

Answer: correction has been made (lines 362 and 364)

Reviewer #3: Studies on chemical pollutants in terrestrial environments are extremely important, especially emerging ones for which there is little or no information in the literature. Bats are mammals with unique characteristics, and therefore, they perform fundamental ecosystem services for the balance of the environment. Precisely performing these services and their proximity to agricultural and industrial areas, they become contaminated with the most diverse chemical pollutants. Taking into account the difficulty in obtaining non-invasive samples to the individual, alternative matrices, such as the use of guano, are welcome.

The manuscript is easy to read and understand. I consider the article suitable for publication in "PLOS ONE". However, some modifications must be taken into consideration

Answer: Thank You for your opinion

Main concerns:

*Add information about benzophenone can increase significantly under favorable temperature and time conditions, probably due to the additional degradation of Octocrylene.

Answer: The information suggested by the Reviewer has been added (Lines 50-53)

*L92: Add information about on the effects of benzophenone on laboratory rats.

Answer: Information about the influence of BPs on laboratory rats has been added (lines 114-118)

*L131: Georeference the map.

Answer: Map (in old version figure 1, and in new ver5sion figure 2) has been reedited. Now it is made by the authors

*L134: Add information about recreational or tourist water areas. The use of UV-filter in these places is higher and can influence contamination.

Answer: Information suggested by the reviewer has been added in table 1 (line 169)

*L220: Rephrase the figures, especially 2 and 3.

Answer: Figure legends has been reedited and clarified – figure caption (lines 257-261)

*The study needs to discuss more about the species' characteristics, such as food guild, foraging areas and behavior, since the bioaccumulation rate can be affected by these topics. Furthermore, it is important to clarify some intraspecific information on the species based on the chosen matrix in the study.

Answer: Information about biology, behavior and feeding of bat species included into the study has been added (lines 305-322)

Minor concerns:

*L62: Add information about carcinogenic potential.

Answer: Information suggested by the Reviewer has been added (Lines 79-80)

*L87: Add current data on benzophenone-3 in zebrafish.

Answer: Information suggested by the Reviewer has been added (lines 102-107)

*L88: Add article "Comparative acute toxicity of benzophenone derivatives and bisphenol analogues in the Asian clam Corbicula fluminea."

Answer: Article has been added (line 108, reference no. 52)

*L88: Add article about freshwater planarian Dugesia japonica

Answer: Article has been added (line 109, reference no. 54)

*L144: As this is an unprecedented work, all details are important for future replications. Therefore, I suggest better developing how the samples were collected and stored in a way that prevents their accidental extrinsic pollution with BPs.

Answer: information about collecting and storage of the samples has been added (182-183)

*L252: "Okereke and Abdel-Rhaman 1994; Okereke et al. 1994"

Answer: The error has been corrected (reference no. 86, 87)

*L280: "Okereke"

Answer: The error has been corrected (reference no. 86)

According to comments placed in the manuscript texts. All suggestions of the reviewer concerning the adding new references have been taken into account.

According table 3: 

1) Of course, if the reviewer deems it necessary, the authors may include the table in the supplementary materials, but in the authors' opinion, the table in the main body of the manuscript makes it easier for the reader to understand the text without the need to additionally search for a table in supplementary materials.

2) According to the suggestion of the reviewer, information about study on bat guano has been moved to the last row of the table (table 3 – line 269)

According differences in BP-1 levels between colonies 3 and 4, which are located relatively close to each other additional information about characterization of colonies has been added (lines 66-371)

Editor comments

1. New version of the manuscript meets PLOS ONE's style requirements

2. In view of the Act for the Protection of Animals for Scientific or Educational Purposes of 15 January 2015 (Official Gazette 2015, No. 266), applicable in the Republic of Poland, the agreement of Ethical Committee was not required for activities carried out during the experiment. Guano samples were not invasive for animals. Moreover samples were collect in such a way that not to scare or stress the animals. Building administrators and persons responsible for the protection of specific bat colonies gave verbal consent to collect samples. Such information has been placed in the “materials and methods” – lines 177-182

3. Funding information has been deleted from the manuscript

4. Information about ethic statement (see point 2) was placed in “material and methods” and has been deleted from other parts of the manuscript

5. The map (figure 1 in old version of the manuscript and figure 2 in new version) has been replaced by the schematic map prepared by the authors and therefore does not need to be copyrighted

6. Figures have been reedited and properly captioned

7. Captions for Supporting Information files have been included at the end of the manuscript

The authors hope that mentioned above corrections will allow to publish the manuscript in PLOS ONE.

---

## [Editor Report · Decision Letter 1]

22 Mar 2024

Biomonitoring of benzophenones in guano samples of wild bats in Poland

PONE-D-23-43946R1

Dear Dr. Rytel,

We’re pleased to inform you that your manuscript has been judged scientifically suitable for publication and will be formally accepted for publication once it meets all outstanding technical requirements.

Kind regards,

Lyi Mingyang, Ph.D.

Academic Editor

PLOS ONE

Additional Editor Comments (optional):

Dear Authors,

Thank you very much for sending us this new and improved version of the manuscript.

I have personally read this R1 version and found it suitable for publication in PLOSONE.

Best Regards,

LM
---

## [Editor Report · Acceptance letter]

27 Mar 2024

PONE-D-23-43946R1 

PLOS ONE

Dear Dr. Rytel, 

I'm pleased to inform you that your manuscript has been deemed suitable for publication in PLOS ONE. Congratulations! Your manuscript is now being handed over to our production team.

Kind regards, 

on behalf of

Professor Lyi Mingyang 

Academic Editor

PLOS ONE